# Dietary Supplementation with Hazelnut Oil Reduces Serum Hyperlipidemia and Ameliorates the Progression of Nonalcoholic Fatty Liver Disease in Hamsters Fed a High-Cholesterol Diet

**DOI:** 10.3390/nu11092224

**Published:** 2019-09-14

**Authors:** Jen-Her Lu, Kai Hsia, Chih-Hsun Lin, Chien-Chin Chen, Hsin-Yu Yang, Ming-Huei Lin

**Affiliations:** 1Department of Pediatrics, Taipei Veterans General Hospital, Taipei 11217, Taiwan; hkay1008@gmail.com (K.H.); jessie55556666@gmail.com (H.-Y.Y.); linda850328@gmail.com (M.-H.L.); 2Department of Pediatrics and Surgery, School of Medicine, National Yang-Ming University, Taipei 11221, Taiwan; chlin12@vghtpe.gov.tw; 3Division of Plastic Surgery, Department of Surgery, Taipei Veterans General Hospital, Taipei 11217, Taiwan; 4Department of Pathology, Ditmanson Medical Foundation Chia-Yi Christian Hospital, Chiayi 600, Taiwan; hlmarkc@gmail.com; 5Department of Cosmetic Science, Chia-Nan University of Pharmacy and Science, Tainan 717, Taiwan

**Keywords:** hazelnuts oil, hyperlipidemia, steatohepatitis, nonalcoholic fatty liver disease, hamster

## Abstract

*Objective*: Hazelnut oil (HO) is rich in monounsaturated fatty acids and polyunsaturated fatty acids. This study intended to analyze the effects of hazelnut oil supplementation on the serum lipid profile and nonalcoholic fatty liver disease in hamsters fed a high-cholesterol (HC) diet. *Methods*: Hamsters were fed a basic diet (control group) and an HC diet (HC group) for 16 weeks (*n* = 10 in each group). Hamsters were fed an HC diet for four weeks to induce hyperlipidemia and were then fed an HC diet enriched with 5% (low-dose HC + HO group; *n* = 10) and 10% HO (high-dose HC + HO group; *n* = 10) for 12 weeks. Serum lipid levels, hepatic changes (including steatosis, inflammation, and fibrosis), and hepatic prooxidant-antioxidant status (malondialdehyde (MDA), superoxide dismutase (SOD), glutathione peroxidase (GPx), and glutathione S-transferase (GST)) were evaluated after the treatment period. *Results*: Hamsters in the control group showed normal serum lipid profiles, normal liver function, and moderate glycogen storage without hepatic steatosis. Hamsters in the HC group showed severe hyperlipidemia, severe hepatic steatosis, and moderate steatohepatitis (mononuclear cell and neutrophil infiltration, oval cell hyperplasia, and fibrosis). Compared to the HC group, both the low-dose and the high-dose HC + HO groups showed a significant reduction of hyperlipidemia (serum triglyceride (TG), total cholesterol (TC), low-density lipoprotein cholesterol (LDL-C), and very-low-density lipoprotein cholesterol (VLDL-C levels)) and improved liver function (serum glutamic-oxaloacetic transaminase (SGOT) and serum glutamic pyruvic transaminase (SGPT)). Additionally, compared to the HC group, intrahepatic triglyceride accumulation (IHTC) was significantly higher in the HC + HO group, while the incidence of steatohepatitis was significantly lower. The intake of the HC diet was associated with a higher level of lipid peroxidation (malondialdehyde, MDA) and a lower concentration of hepatic antioxidant enzymes (SOD, GPx, and GST), and all these factors were partially improved in the low-dose and high-dose HC + HO groups. *Conclusions:* Our findings indicate that the intake of HO reduced serum hyperlipidemia and oxidative stress and ameliorated the progression of nonalcoholic fatty liver disease in hamsters fed a high-cholesterol diet.

## 1. Introduction

Diets rich in saturated fatty acids (FA) can induce metabolic disorders and the development of coronary heart disease and nonalcoholic fatty liver disease [1,2]. Dietary FA and excess cholesterol have significant impacts on plasma lipid profiles and are strongly correlated with liver injury [3,4]. Dietary FA content may cause significant changes in the lipoprotein synthesis and lipid composition of cellular structures [5,6,7]. High levels of monounsaturated FA (MUFA) and/or polyunsaturated FA (PUFA) have hypolipidemic effects [8,9,10,11,12]. Hazelnut oil (HO) is rich in MUFA and antioxidants (vitamin E, phytosterol, vitamin B6, etc.). A hazelnut-rich diet may decrease the atherogenic tendency of low-density lipoprotein cholesterol (LDL-C) by lowering the susceptibility of LDL-C to oxidation [13]. Furthermore, the consumption of different forms of hazelnuts enriched in unsaturated FA (UFA) can decrease the levels of serum total triglycerides (TG), total cholesterol (TC), and LDL-C induced by saturated FA (SFA) [14,15,16,17,18].

Serum high-density lipoprotein cholesterol (HDL-C) is an anti-atherogenic lipoprotein which plays a role in reversing cholesterol transport from peripheral tissues to the liver. Low HDL-C levels induce the development of coronary heart disease and cerebrovascular disease. Dietary UFA ingestion has been shown to be associated with increased serum HDL-C levels in animals and humans [19,20,21]. One of the most intriguing impacts of dietary UFA on the serum lipid profile is its effect on hepatic lipid metabolism [3,22]. Considering that diets appropriately enriched in HO could be used to correct serum and hepatic lipid metabolism, this study aimed to determine the effects of HO in adult hamsters fed with HC diets. The serum, hepatic, and fecal lipid profiles were measured. Furthermore, histopathological analysis was carried out on hamster livers to evaluate the changes in hepatic steatosis, inflammation, and fibrosis. 

## 2. Materials and Methods 

### 2.1. Animals 

A total of 40 five-week old male hamsters were used in the study. The animals were acquired from the Animal Center of the Academia Sinica, Taiwan, and were singly housed in separate boxes placed in a room at a temperature of 22 °C and under a 12 h light/dark cycle. The animals had free access to food and water and were allowed to acclimatize for one week prior to the start of the study. All protocols used in this study were accepted by the ethical committee of the National Yang-Ming University, Taiwan (ICUC Nr. 201805). This study carefully followed the ethical guidelines of the National Research Council Guide for the Care and Use of Laboratory Animals (1996).

### 2.2. Analytical Determination of Supplemented Dietary Fat

Hazelnut oil was purchased from a local manufacturer, and fatty acid methyl esters (FAMEs) from the oil samples were prepared as described by Issaoui et al. [23]. Individual FAMEs were separated and quantified by gas chromatography using a Model 5890 Series II instrument (Hewlett-Packard, Palo Alto, CA, USA) equipped with a flame ionization detector, and a DB-23 fused silica capillary column (60 m length, 0.32 mm i.d., 0.25 μm film thickness; HP-Agilent Technologies, Wilmington, DE, USA). Table 1 shows the fatty acid composition of the hazelnut oil used in this study.

### 2.3. Experimental Design 

The animals were randomly assigned into four groups (*n* = 10 in each group) and subjected to the following different dietary regimens: (a) basic diet control group: 10 hamsters were fed with commercial LabDiet Rodent 5001 rodent feed (Labdiet, St. Louis, MO, USA) for 16 weeks; (b) HC group: 10 hamsters received regular feed enriched with an HC diet (0.2% cholesterol, 0.2% bile salt, 10% egg yolk powder, and 10% lard) for 16 weeks; (c) low-dose and high-dose HC + HO groups (*n* = 10 in each group): hamsters were fed with an HC diet for four weeks to induce hyperlipidemia and were then fed for 12 weeks (5th week to 16th week) with HC plus 5% HO (low-dose HC + HO group) and HC plus 10% HO (high-dose HC + HO group) diets. All diets were stored at −20 °C and fresh food was provided to the hamsters every two days. Diets were prepared in pellet form and oils were manually applied to the pellets. The compositions of the diets are listed in Table 2. The animals’ growth and food consumption were monitored every working day by recording body weight. 

### 2.4. Sample Collection and Preparation 

Blood was collected from all animals via periorbital puncture at the start day of the study (Initial), at 5 weeks (WK0), 9 weeks (WK4), 13 weeks (WK8), and at the end of 16 (WK12) weeks. Twenty-four h after the last day of treatment, the animals were anesthetized after an overnight fast (16 h). Blood samples were transferred into anticoagulant-free vials and allowed to stand for 30 min to clot. Afterwards, the vials were centrifuged at 300 g for 10 min and the resultant serum was used for further analysis. At the end of the feeding time (16 weeks), the animals were fasted overnight and anesthetized with sodium pentobarbital (50 mg/kg, i.p.). Blood was collected in tubes containing EDTA by cardiac puncture. Plasma samples were obtained by centrifugation and stored at −70 °C until analysis. The livers were rapidly removed, washed in 0.9% NaCl, and kept on ice. Portions of the liver were fixed in 10% buffered formalin for 24 h, dehydrated in a gradual series of alcohols and diaphanous in xylene for paraffin embedding. Paraffin blocks were sectioned at 4 μm and stained with hematoxylin and eosin for histological examination by optical microscopy.

### 2.5. Determination of Serum Lipid and Lipoprotein Cholesterol

Blood samples were collected via non-heparinized capillary tubes into Eppendorf tubes. The whole blood was left at room temperature for 30 min and serum was then harvested after centrifugation at 1500× *g* at 4 °C for 20 min. Concentrations of serum TC, TG, HDL-C, LDL-C, GOT (glutamic-oxaloacetic transaminase), and GPT (glutamic pyruvic transaminase) were measured by a TOSHIBA-C16000 automated clinical chemical analyzer (Toshiba Corporation, Tokyo, Japan) using commercial kit for TC, HDL-C (HDL-EX), LDL-C (HDL-EX; Denka Seiken Co., Tokyo, Japan), GOT (GOT-JS; Denka Seiken Co., Tokyo, Japan), GPT (GPT; Denka Seiko Co., Tokyo, Japan), and TG (triglycerides liquid; Sentinel CH SpA, Milan, Italy). The very-low-density lipoprotein cholesterol (VLDL-C) value in this experiment was calculated by the formula VLDL-C = TC − (LDL-C + HDL-C).

### 2.6. Hepatic and Fecal Lipid Analysis

Liver lipids were extracted according to the method of Folch [24]. A total of 2 g of liver tissue and 1 g of lyophilized feces were homogenized with chloroform/methanol (2/1, *v*/*v*) to a final volume of 20 times the volume of the tissue sample (1 g in 20 mL of solvent mixture) in an ice bath. The homogenate was filtered using Whatman No. 1 filter paper to obtain the liquid phase, whose volume was then replenished to 10 mL. The extract was dried under N_2_ and resuspended in isopropanol. Liver and fecal TC and TG levels were measured using a TOSHIBA-C16000 automated clinical chemical analyzer and the aforementioned commercial kits. 

### 2.7. Liver Antioxidant and Paroxidant Analysis

Liver portions were homogenized in ice-cold 0.15 M KCL (10%, *w*/*w*). Lipids were then extracted with chloroform:methanol (2:1). After extraction and evaporation, hepatic livers were re-dissolved in isopropanol and hepatic cholesterol and triglyceride levels were assayed by SGPT (serum glutamic pyruvic transaminase) Activity Assay Kit and AST (aspartate aminiotransferase) Activity Assay Kit (Sigma-Aldrich, Darmstadt, Germany). The level of malondialdehyde (MDA) in the liver was assessed by the thiobarbituric acid test (thiobarbituric acid reactive substances (TBARS) Assay Kit; Cayman Chemical Company, Ann Arbor, MI, USA). The breakdown product of 1,1,3, 3-tetraethoxypropane was used as a standard. Hepatic superoxide dismutase (SOD) activity was assayed by its ability to increase the effect of riboflavin-sensitized photooxidation of orthodianisidine in postmitochondrial fractions (Superoxide Dismutase Assay Kit; Cayman Chemical Company). Glutathione peroxidase (GSH-Px) and glutathione transferase (GST) activities were measured using cumene hydroperoxide and 1-chloro-2,4-dinitrobenzed as substrates, respectively, in postmitochondrial fractions. Protein levels were determined using bicinchoninic acid. (Glutathione Peroxidase and Glutathione S-Transferase Assay Kit; Cayman Chemical Company, Ann Arbor, MI, USA).

### 2.8. Histopathological Analysis

Livers were fixed with 10% formalin for 24 h before being cut into pieces with a length of 4 μm for paraffin sections and being stained with hematoxylin-eosin (HE) reagent. The degree of lesions stained with HE was graded from one to five depending on severity according to the scoring system from Shackelford et al.: 1 = minimal (<1% lesions); 2 = slight (1–25%); 3 = moderate (26–50%); 4 = moderate/severe (51–75%); 5 = severe/high (76–100%) [25]. 

### 2.9. Statistical Analysis 

The results were expressed as mean ± SD. The data were analyzed by two-way ANOVA, followed by the Kruskal–Wallis test when appropriate, using Prism software (GraphPad Software Inc., San Diego, CA, USA). The histopathological results were analyzed by Kruskal–Wallis, Wilcoxon, and Mann–Whitney U tests. The Cochran Q test was used to determine the location in the acinar hepatic zones. A *p*-value < 0.05 was considered to indicate a statistically significant difference among the groups. 

## 3. Results

### 3.1. Food Ingestion and Weight Gain

Different treatment regimens were associated with significantly different weight gain and food consumption among the hamsters (Table 3). The food consumption of the basic diet control group remained constant, with the animals exhibiting steady weight gain. The food consumption of the HC group remained constant, with the animals exhibiting less weight gain than the control group. Compared to the HC group, food consumption was significantly lower in the two HC + HO groups. Additionally, compared to the HC group, the two HC + HO groups exhibited significantly lower weight gain (13.9%). Compared to the control group, the feed efficacy was significantly lower in the HC group and slightly higher in the high-dose HC + HO group.

### 3.2. Serum Lipid Profile

Compared to the control group, the serum levels of TC, TG, LDL-C, and VLDL-C were significantly higher in the HC group. Additionally, after four weeks of the high-cholesterol diet, the serum lipid profiles of the two HC + HO groups were significantly higher than that of the HC group. Furthermore, compared to the HC group, the serum lipid profiles (TG, TC, LDL-C, and VLDL-C) and LDL-C/HDL-C ratios of the low-dose and high-dose HC + HO groups were significantly lower at 4, 8, and 12 weeks (Table 4). Compared to the lipid profile in the HC group, both the low-dose and high-dose HC + HO groups had significantly higher levels of HDL-C. 

### 3.3. Liver Function 

Compared to the control group, the liver functions (serum glutamic-oxaloacetic transaminase (SGOT) and serum glutamic pyruvic transaminase (SGPT)) in the HC group were significantly impaired (Table 5). A significant improvement in liver function was observed at 4, 8, and 12 weeks after the ingestion of the low-dose and high-dose HC + HO diets. 

### 3.4. Hepatic Antioxidant and Prooxidant Analysis

Compared to the control group, MDA levels were significantly higher in the HC group and remained unchanged in the low-dose and high-dose HC + HO groups. Additionally, compared to the control group, SOD levels were significantly lower in the HC group. Furthermore, compared to the HC group, liver SOD levels were significantly higher in both the low-dose and high-dose HC + HO groups (Figure 1).

Compared to the control group, the level of glutathione peroxidase (GPx) activity was significantly lower, and the GST level was significantly higher, in the HC group and the low-dose and high-dose HC + HO groups.

### 3.5. Liver Histopathology

#### 3.5.1. Microvesicular Steatosis 

Liver samples from the control group had no microvesicular steatosis (Figure 2). However, liver samples from the HC group showed severe/high microvesicular steatosis in hepatocytes cytoplasm. Additionally, compared to the HC group, the low-dose and high-dose HC + HO groups had a mildly increased level of microvesicular steatosis. 

#### 3.5.2. Macrovesicular Steatosis 

Liver samples from the control group showed no macrovesicular steatosis. Only minimal macrovesicular steatosis was found in the HC group. Hamsters in the low-dose and high-dose HC + HO groups had significantly higher levels of macrovesicular steatosis.

#### 3.5.3. Inflammatory Changes 

Liver samples from the control group had no inflammatory cell infiltration. However, liver samples from the HC group had significantly higher mononuclear cell and neutrophil infiltration. Additionally, liver samples from the two HC + HO diet groups showed significantly lower levels of neutrophil infiltration. The HC diet was found to activate Kupffer cells in the liver. The activation of residual Kupffer cells suggested the presence of toxic lipid compounds in adjacent hepatocytes and/or in the Kupffer cells themselves. Moreover, a significant reduction of mononuclear cell infiltration and neutrophil infiltration, and a moderate activation of residual Kupffer cells, was observed in both HC + HO groups (Table 6).

#### 3.5.4. Fibrotic Changes 

Liver specimens in the control group had no fibrotic changes. Liver specimens in the HC group showed significant fibrotic changes, including minimal oval cell hyperplasia, hepatic cell necrosis, minimal fibrosis, and moderate glycogen accumulation (Table 6). Significantly less severe fibrotic changes were found in both HC + HO groups.

#### 3.5.5. Glycogen Accumulation 

Liver specimens from the control group had moderate glycogen accumulation. Liver specimens from the HC group showed significantly lower glycogen accumulation. Significantly lower glycogen accumulation was observed in both HC + HO groups. 

### 3.6. Lipid Contents in Liver Parenchymal and Fecal Material

Compared to the control group, the TC and TG contents were significantly higher in the livers from the HC group (Table 7). Compared to the HC group, the TC contents in the livers from the two HC + HO groups were unchanged, however, the TG contents were significantly higher. No significant differences were observed in the fecal lipid levels of TC and TG in the four different groups, however, mildly elevated fecal lipid levels of TC and TG were observed in both HC + HO groups. 

## 4. Discussion

Hazelnut is a fatty food and its regular consumption can theoretically be expected to lead to body weight gain. In this study, the hamsters in the control group showed steady weight gain. However, in the HC group and the high-dose HC + HO group, no significant changes in body weight were observed throughout the study period. Significantly lower body weight was observed in the low-dose HC + HO diet group. This result is concordant with the results of other investigators who observed an inverse relationship between frequent nut consumption and body mass index (BMI) [26,27].

### 4.1. Serum Lipid Profile in the HC + HO Groups 

In this study, we used hamsters fed with a high-cholesterol diet as a model to investigate the effects of hazelnut consumption on blood lipid profiles in hamsters with hyperlipidemia. It was found that diets containing different doses of hazelnut oil were associated with lower levels of serum TC and LDL-C and higher levels of serum HDL-C. The main reason for this effect on serum cholesterol profiles could be the unique lipid content of hazelnuts. The hazelnut oil used in this study contains a high content of MUFA (77%), mainly consisting of oleic acid (18:1), and low contents of PUFA (17%) and saturated fatty acids (8%). This lipid content can explain most of the lowering of serum LDL-C levels, as has previously been observed for olive oil. Oleic acid has been clearly shown to reduce serum cholesterol levels [28]. The additional micronutrients and bioactive substances present in hazelnut oil—including vitamin E (45 mg/100 mg), phytosterols, *L*-arginine, polyphenols, and folate—further reduce serum hyperlipidemia. The reduction of serum lipid profile caused by hazelnut consumption is similar to that found for a variety of other nuts [29,30]. Encouraging results have been obtained regarding the benefits of diets rich in unsaturated fatty acids [5,11,12,31,32]. Previous studies revealed that a diet rich in cholesterol and oleic acid reduced serum LDL-C levels in hamsters [33,34]. Additionally, a previous study also showed that MUFA-containing oils had beneficial effects for lipids and lipid peroxidation [35]. The effect of HO consumption on serum lipids and lipid peroxides has not been clearly established [16,17,28,35]. The steady-state balance of hepatic triglycerides is controlled by the consumption of fatty acids via mitochondrial beta-oxidation [36]. Previous studies have found that the daily dose and duration of dietary hazelnut intervention is crucial for producing a significant reduction of serum levels of TG, LDL-C, VLDL-C, and TG [37,38]. Our results suggest that the antioxidant effect of hazelnut oil on the liver also plays an important role. This study revealed that low-dose (5%) hazelnut oil supplementation in hamsters, equivalent to the daily required amount in humans of 25 g, was able to ameliorate the change in serum lipid profile induced by cholesterol-rich diets within four weeks. This study also found that the reduction of serum hyperlipidemia caused by dietary HO was not dose dependent, and that excess intake of HO did not have any further effect on serum lipid profiles. 

### 4.2. Hepatic Steatosis in the HC + HO Groups

Hepatic steatosis is related to intrahepatic triglyceride (IHTG) accumulation. Liver steatosis consists of micro- and macrovesicular steatosis and results in an imbalance between hepatic triglyceride storage and lipid turnover [39]. Microvesicular steatosis is defined as a centrally located nucleus with the cytoplasm replaced by bubbles of fat [30]. Macrovesicular steatosis is defined as severe steatosis that is distinguished by the presence of large-droplet steatosis, which involves the presence of bubbles of fat in the cytoplasm with displacement of the nucleus to the edge of the cell. Macrovesicular steatosis alone is considered to have a good long-term prognosis, with rare progression to fibrosis or cirrhosis [40]. In our study, the HC diet was found to be associated with severe and diffused microvesicular steatosis with minimal macrovesicular steatosis. Dietary supplementation with a low dose (5%) of hazelnut oil did not change the status of hepatic steatosis, however supplementation with a high dose (10%) of hazelnut oil increased the degree of macrovesicular steatosis. Dietary supplementation with monosaturated fatty acids has been found to have beneficial effects for organisms, however it has also been found to concomitantly increase the levels of hepatic lipids, especially triglycerides [41,42,43,44]. It has been proposed that an increased level of intrahepatic triglyceride accumulation is a biomarker for protection against liver damage including inflammatory changes and fibrosis [45,46]. 

### 4.3. Steatohepatitis in the HC + HO Groups

Chronic injury conditions in the liver are usually associated with the induction of inflammation, while lymphocytes and inflammatory responses have also been suggested to play a role. The composition of fatty acids delivered to and stored within the liver is an independent risk factor for progression to nonalcoholic steatosis hepatitis (NASH) [47]. In this study, hamsters showed abnormal liver function following four weeks of HC diet ingestion. Liver function impairment improved significantly following four weeks of HC + HO diet ingestion. Kupffer cells play an important role in the pathogenesis of inflammatory liver diseases leading to fibrosis [48,49]. Toxic lipid compounds can activate resident macrophages (Kupffer cells) and recruit blood-derived monocytes and neutrophils, which have been identified as key elements for the initiation and progression of hepatitis. In the present study, histopathological findings in the HC group revealed slight neutrophil infiltration of mild inflammatory and fibrotic changes and interference with the accumulation of glycogen in the liver. Oval cells are used specifically in rodents, however cells with similar characteristics have been reported in human nonalcoholic fatty liver disease [50]. In humans, these cells are usually referred to as hepatic progenitor cells or intermediate hepatic biliary cells. Oval cells are related to hepatic regeneration and are activated by different hepatic injuries. In this study, it was found that oval cell hyperplasia was significantly higher in the HC group and significantly lower in the high-dose HC + HO group, which is concordant with the attenuated toxic effect of saturated fatty acids. 

The hallmarks of nonalcoholic fatty liver disease are impaired suppression of hepatic glycogen production and increased intrahepatic triglyceride content [50]. In this study, we found that hepatic glycogenesis was severely depressed in the HC group and significantly increased in both of the HC + HO groups. Our results suggest that, although hazelnut oil increased hepatic lipid accumulation, it nevertheless normalized hepatic metabolic function and glycogenesis dysfunction. The oxidative stress, lipotoxicity, and inflammation of saturated fatty acids play a key role in the progression of hepatic steatosis to steatohepatitis [51,52]. The action of superoxide dismutase on the secondary breakdown products of oxidative stress is related to the activity of the antioxidant defense system. Glutathione peroxidase is related to the antioxidant activity and detoxication ability [15,53]. In our study, the HC diet was found to be associated with oxidative stress in hamsters, leading to significant decreases in the levels of the antioxidant SOD and GPx, and a significant increase in the levels of the prooxidant MDA, in the liver. It was also found that abnormal prooxidant-antioxidant levels can be significantly converted by the concomitant ingestion of hazelnut oil.

## 5. Conclusions

The results of our study indicate that the HC diet contributes to hypercholesterolemia, hepatic steatosis, and steatohepatitis. The two HC + HO diets effectively reduced the serum levels of LDL-C, VLDL-C, and TC, which may be due to the recovery of hepatic antioxidant function. Despite the aggregation of hepatic steatosis in hamsters fed a low-dose or high-dose HC + HO diet, inflammatory and fibrotic changes were significantly diminished with normalized hepatic glycogenesis.

## 6. Limitation and Future Works

Compared to other groups, the animals from the low-dose HC + HO group had a significant low concentration of TG and high concentration of LDL-C in the initial stage, which is probably due to the wide normal range of serum lipids in hamsters. Although the animals in the low-dose HC + HO group successfully induced hyperlipidemia after feeding with an HC diet for 4 weeks at the same starting point (5th week) as other groups, this should be considered as a limitation of this experiment. 

This experiment focused on the efficacy of dietary supplementation with different doses of HC + HO diet to reduce serum hyperlipidemia and fatty liver disease in hamsters with a high-cholesterol diet. The effect of HO supplementation with normal diet on blood lipids and fatty liver disease remains unclear. Further studies using different doses of HO with a normal diet might reveal the effect of HO itself on serum lipid profile and prevention of nonalcoholic fatty liver disease (NAFLD) to NASH. The efficacy of HO can be further evaluated by comparison studies with other dietary oils such as olive oil (rich in MUFA) and soybean oil (rich in PUFA).

## Figures and Tables

**Figure 1 nutrients-11-02224-f001:**
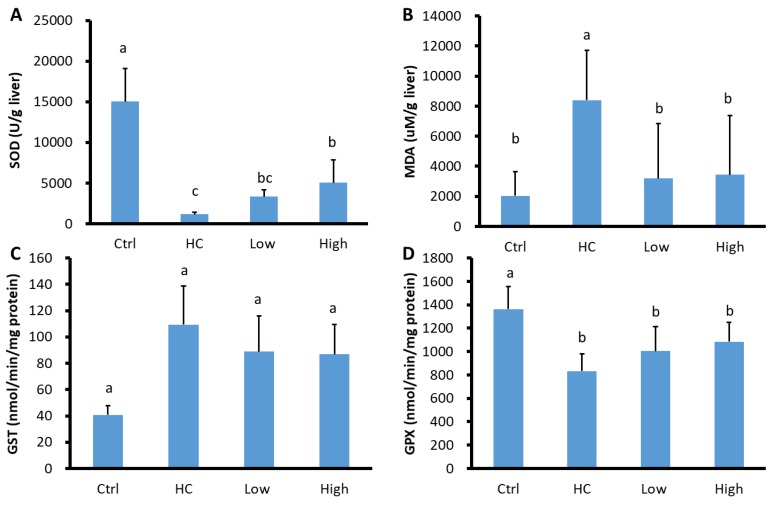
Hepatic prooxidant-antioxidant status in different groups. (**A**) Antioxidant superoxide dismutase (SOD) level; (**B**) the prooxidant index level of malondialdehyde (MDA); (**C**) the antioxidant glutathione S-transferase (GST) level; (**D**) the antioxidant glutathione peroxidase (GPx) level. Ctrl: control group; HC: HC group; Low: low-dose HC + HO group; High: high-dose HC + HO group. Letters a, b, and c indicate that a significant difference was detected by one-way ANOVA with Duncan’s multiple range test.

**Figure 2 nutrients-11-02224-f002:**
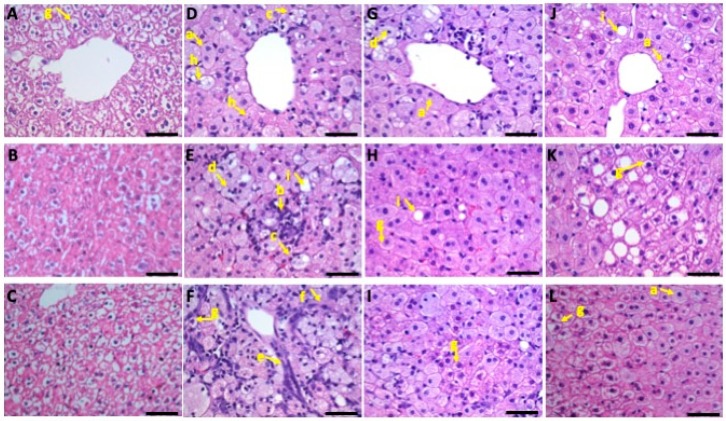
Control group (**A**–**C**): the liver showed no microvesicular or macrovesicular steatosis and no infiltration, however it showed severe/high glycogen accumulation (arrow **g**). HC group (**D**–**F**): the liver presented severe/high microvesicular fatty change (arrow **a**), moderate infiltration of mononuclear cells (arrow **b**) and neutrophils (arrow **c**), moderate Kupffer cell hyperplasia (arrow **d**), moderate oval cell hyperplasia (arrow **e**), moderate fibrosis (arrow **f**), slight glycogen accumulation (arrow **g**), and minimal hepatocellular necrosis (arrow **h**). Low-dose HC + HO group (**G**–**I**): the liver presented severe/high microvesicular fatty change (arrow **a**), minimal macrovesicular fatty change (arrow **i**), slight infiltration of mononuclear cells (arrow **b**) and neutrophils (arrow **c**), moderate Kupffer cell hyperplasia (arrow **d**), slight oval cell hyperplasia (arrow **e**), slight fibrosis (arrow **f**), minimal glycogen accumulation (arrow **g**), and minimal hepatocellular necrosis (arrow **h**). High-dose HC + HO group (**J**–**L**): the liver presented severe/high microvesicular fatty change (arrow **a**), slight macrovesicular fatty change (arrow **i**), slight infiltration of mononuclear cells (arrow **b**) and neutrophils (arrow **c**), slight Kupffer cell hyperplasia (arrow **d**), slight oval cell hyperplasia (arrow **e**), slight fibrosis (arrow **f**), slight glycogen accumulation (arrow **g**), and minimal hepatocellular necrosis (arrow **h**). Scale bars are 200 µm for (**A**,**D**,**G**,**J**); and 50 µm for (**B**,**C**,**E**,**F**,**H**,**I**,**K**) and (**E**,**H**,**L**).

**Table 1 nutrients-11-02224-t001:** Fatty acid composition of hazelnut oil.

Type	Fatty Acid Name	Composition	Percentage (%)	Amount (%)
Saturated Fatty Acid	Myristic Acid	C14:0	0.03	7.52
	Pentadecanoic Acid	C15:0	0.01	
	Palmitic Acid	C16:0	4.96	
	Margaric Acid	C17:0	0.04	
	Stearic Acid	C18:0	2.46	
	Lignoceric Acid	C24:0	0.01	
Monounsaturated Fatty Acid	Oleic Acid (ω-9)	C18:1	80.6	80.9
	Palmitoleic Acid	C16:1	0.15	
	Gadoleic Acid	C20:1	0.19	
Polyunsaturated Fatty Acid	Linoleic Acid (ω-6)	C18:2	11.2	11.5
	Linolenic Acid (ω-3)	C18:3	0.27	
Trans Fat	Conjugated Linolenic Acid Trans-Octadecatrienoic acid	Trans C18:2 Trans C18:3	0.13	0.13

**Table 2 nutrients-11-02224-t002:** Nutrients composition of the control, high cholesterol, and hazelnut oil-enriched diets.

Per 100 g	Control	HC	Low Dose HC + HO	High Dose HC + HO
Energy (Kcal)	409	480	496	513
Protein (g)	29.39	26.8	25.3	23.8
Total Fat (g)	5.78	20.1	23.9	27.7
Saturated Fatty Acid (g)	1.22	6.6	7	7.4
Unsaturated Fatty Acid (g)	2.89	2.3	5.9	9.6
Monounsaturated Fatty Acid (g)	1.2	1	4.2	7.5
Oleic Acid (g)	0	0	3.3	6.6
Polyunsaturated Fatty Acid (g)	1.69	1.3	1.7	2.1
Linolenic Acid (g)	1.54	1.2	1.6	2
Alpha-Linolenic acid (g)	0.13	0.1	0.1	0.1
Cholic Acid (g)	0.22	0.38	0.36	0.35
Cholesterol (g)	0.03	0.22	0.22	0.22
Phytochemicals (g)	0	0	0.005	0.01
Carbohydrate (g)	59.84	48.04	45.04	42.05
Fiber (g)	4.1	3.26	3.06	2.85
Sodium (mg)	260	219	206	193
Vitamin E (mg-αTE)	4.1	3.2	4.7	6.2

Control: basic diet control group; HC: high-cholesterol diet group; low-dose HC + HO: high-cholesterol diet enriched with a low dose (5%) of hazelnut oil; high-dose HC + HO: high-cholesterol diet enriched with a high dose (10%) of hazelnut oil. HC: high cholesterol. HO: Hazelnut oil.

**Table 3 nutrients-11-02224-t003:** Body weight and food ingestion of each group.

	Group	Initial	WK0	WK4	WK8	WK12
**Body weight** **(g)**	Control	115.04 ± 6.13	140.05 ± 12.81 ^a^	157.59±15.56 ^a^	170.43±19.54 ^a^	175.08 ± 17.66 ^a^
	HC	120.16 ± 6.04	135.25 ± 5.51 ^a^	152.99±9.39 ^a^	154.44±8.00 ^a,b^	154.82 ± 9.78 ^b^
	Low dose HO + HC	114.6 ± 12.2	143.93 ± 8.42 ^a^	144.62 ± 10.88 ^b^	147.70 ± 12.65 ^b^	144.55 ± 11.38 ^b^
	High dose HO + HC	128.9 ± 7.6	150.05 ± 10.52 ^a^	154.84 ± 10.98 ^a,b^	175.61 ± 13.69 ^a^	184.26 ± 13.83 ^a^
Ingestion of food (g)	Control	7.33 ± 0.95	8.43±0.62 ^b^	7.45±0.71 ^a^^,^^b^	7.77±0.89 ^a^	7.56±0.68 ^a^
	HC	8.13 ± 0.13	9.62 ± 0.66 ^b^	8.08 ± 0.86 ^a^	7.56 ± 0.92 ^a^	7.48 ± 1.03 ^a^
	Low dose HO + HC	10.3 ± 1.39	11.84 ± 0.50 ^a^	6.67 ± 0.51 ^b^	6.49 ± 0.39 ^b^	5.29 ± 0.73 ^a,b^
	High dose HO + HC	9.2 ± 0.07	11.81 ± 0.47 ^a, b^	4.87 ± 1.77 ^b^	7.25 ± 1.01 ^a,b^	6.26 ± 0.44 ^a^
Feed efficiency	Control		2.97 ± 1.16	2.36 ± 0.72	1.65 ± 0.58	0.62 ± 0.52
	HC		1.57 ± 0.61	2.20 ± 1.18	0.19 ± 1.16	0.05 ± 0.42
	Low dose HO + HC		2.55 ± 1.07	0.11 ± 0.77	0.46 ± 0.57	-0.57 ± 0.46
	High dose HO + HC		1.79 ± 0.49	1.10 ± 1.34	2.78 ± 1.06	1.37 ± 0.73

Data are expressed as mean ± SD (*n* = 10 hamsters per group). Control: basic diet control group; HC: high-cholesterol diet group; low-dose HC + HO: high-cholesterol diet enriched with a low dose (5%) of HO; high-dose HC + HO: high-cholesterol diet enriched with a high dose (10%) of HO; WK: week. Comparisons between groups were made using Duncan’s multiple range test. Superscript letters (^a,b^) indicate significant differences between groups (*p* < 0.05).

**Table 4 nutrients-11-02224-t004:** Serum lipid profile in different groups.

	Group	Initial	WK0	WK4	WK8	WK12
**TC (mg/dL)**	Control	148.00 ± 20.67	108.50 ± 22.83	117.80 ± 22.07	88.50 ± 5.91	95.90 ± 7.11
	HC	143.3 ± 18.87	595.00 ± 158.41 ^b^	749.60 ± 133.07 ^a^	729.20 ± 116.32 ^a^	700.20 ± 106.26 ^a^
	Low dose HC + HO	146.3 ± 12.38	885.20 ± 245.06 ^a^	395.10 ± 125.29 ^b^	336.30 ± 94.18 ^b^	451.80 ± 127.56 ^b^
	High dose HC + HO	144.2 ± 17.61	1028.00 ± 327.41 ^a^	279.50 ± 30.87 ^c^	339.30 ± 58.23 ^b^	400.20 ± 83.65 ^b^
**TG (mg/dL)**	Control	186.60 ± 76.09	154.20 ± 51.23	142.70 ± 63.64	116.50 ± 48.72	151.10 ± 40.22
	HC	142.90 ± 49.48	223.4 ± 27.7794	971.7 ± 146.7485	576.2 ± 221.73	475.9 ± 190.9092
	Low-dose HC + HO	81.99 ± 34.09	1008.50 ± 496.96 ^a^	371.20 ± 252.58 ^a,b^	262.10 ± 163.20 ^c^	224.50 ± 142.73 ^b^
	High-dose HC + HO	127.37 ± 98.72	1474.20 ± 734.42 ^a^	313.40 ± 65.71 ^b^	384.40 ± 59.45 ^b^	363.80 ± 141.04 ^a^
**LDL-C (mg/dL)**	Control	20.31 ± 7.87	14.29 ± 6.72	19.64 ± 6.99	9.85 ± 1.43	13.70 ± 3.04
	HC	18.70 ± 4.73	205.80 ± 90.13 ^a^	229.58 ± 40.06 ^a^	228.83 ± 37.68 ^a^	223.92 ± 38.06 ^a^
	Low-dose HC + HO	41.30 ± 7.70	239.20 ± 72.79 ^a^	75.60 ± 44.96 ^b^	59.67 ± 33.05 ^b^	117.80 ± 60.29 ^b^
	High-dose HC + HO	38.80 ± 9.84	302.62 ± 132.52 ^a^	28.73 ± 8.17 ^c^	67.19 ± 24.67 ^b^	91.95 ± 33.36 ^b^
**HDL-C (mg/dL)**	Control	71.91 ± 9.43	59.69 ± 8.03	58.01 ± 4.58	55.00 ± 3.89	53.35 ± 3.83
	HC	74.15 ± 5.60	88.85 ± 5.77 ^a^	86.50 ± 8.47 ^b^	102.29 ± 7.61 ^b^	106.62 ± 5.08 ^b^
	Low-dose HC + HO	95.61 ± 8.94	82.71 ± 8.40 ^a,b^	123.47 ± 11.35 ^a^	124.27 ± 12.09 ^a^	119.11 ± 11.37 ^a^
	High-dose HC + HO	89.39 ± 8.51	79.39 ± 8.64 ^b^	131.00 ± 11.08 ^a^	117.89 ± 12.26 ^a^	117.90 ± 11.22 ^a^
**LDL-C/HDL-C Ratio**	Control	0.28 ± 0.83	0.24 ± 0.84 ^c^	0.34 ± 0.53	0.18 ± 0.37 ^b^	0.26 ± 0.79 ^b^
	HC	0.25 ± 0.84	2.35 ± 1.14 ^b^	2.70 ± 0.64 ^a^	2.26 ± 0.49 ^a^	2.10 ± 0.38 ^a^
	Low-dose HC + HO	0.43 ± 0.86	2.90 ± 1.14 ^a,b^	0.62 ± 0.37 ^b^	0.49 ± 0.28 ^b^	1.01 ± 0.55 ^b^
	High-dose HC + HO	0.43 ± 1.16	3.98 ± 2.10 ^a^	0.22 ± 0.07 ^c^	0.58 ± 0.24 ^b^	0.79 ± 0.32 ^b^
**VLDL-C**	Control	55.78 ± 14.95	34.52 ± 10.99 ^c^	40.15 ± 14.03 ^c^	23.65 ± 3.65 ^c^	28.85 ± 3.77 ^c^
	HC	50.46 ± 11.06	300.35 ± 116.30 ^a,b^	433.52 ± 102.25 ^a^	398.08 ± 86.03 ^a^	369.66 ± 72.25 ^a^
	Low-dose HC + HO	9.37 ± 6.27	563.29 ± 179.58 ^a^	196.03 ± 82.08 ^b^	152.36 ± 63.65 ^b^	214.89 ± 72.94 ^b^
	High-dose HC + HO	15.98 ± 13.38	645.99 ± 207.69 ^a^	119.77 ± 28.91 ^b^	154.22 ± 37.22 ^b^	190.35 ± 53.39 ^b^

Data are expressed as mean ± SD (*n* = 10 hamsters per group). Control: basic diet control group; HC: high-cholesterol diet group; low-dose HC + HO: high-cholesterol diet enriched with a low dose (5%) of HO; high-dose HC + HO: high-cholesterol diet enriched with a high dose (10%) of HO; TC: total cholesterol; TG: total triglyceride; LDL-C: low-density lipoprotein cholesterol; HDL-C: high-density lipoprotein cholesterol; WK: week. Comparisons between groups were made using Duncan’s multiple range test. Superscript letters (^a–c^) indicate significant differences between groups (*p* < 0.05).

**Table 5 nutrients-11-02224-t005:** Serum liver enzymes in different groups.

	Group	Initial	WK0	WK4	WK8	WK12
	Control	54.60 ± 10.96	61.10 ± 9.69	63.40 ± 7.81	56.10 ± 6.40	50.50 ± 6.29
**GOT (U/L)**	HC	59.9 ± 24.5	172.90 ± 80.78 ^a^	137.30 ± 36.17 ^a^	113.90 ± 23.36 ^a^	96.70 ± 16.23 ^a^
	Low dose HC + HO	58.5 ± 8.80	90.80 ± 26.55 ^b^	58.90 ± 17.34 ^b^	63.70 ± 18.82 ^b^	71.70 ± 24.84 ^b^
	High dose HC + HO	61.9 ± 12.53	81.00 ± 21.10 ^b^	51.60 ± 13.18 ^b^	58.10 ± 12.42 ^b^	62.60 ± 27.01 ^b,c^
	Control	85.80 ± 40.18	81.00 ± 25.94	84.10 ± 22.44	66.80 ± 13.65	81.90 ± 24.07
**GPT (U/L)**	HC	78.4 ± 21.17	688.00 ± 269.41 ^a^	452.60 ± 142.17 ^a^	379.40 ± 94.72 ^a^	311.00 ± 83.76 ^a^
	Low dose HC + HO	95.3 ± 23.44	349.30 ± 110.07 ^b^	151.50 ± 57.80 ^b^	189.40 ± 73.45 ^b^	226.80 ± 91.77 ^b^
	High dose HC + HO	102.8 ± 28.19	349.70 ± 141.63 ^b^	93.50 ± 20.39 ^b^	168.60 ± 35.15 ^b^	156.80 ± 38.18 ^c^

Data are expressed as mean ± SD (*n* = 10 hamsters per group). Control: basic diet control group; HC: high-cholesterol diet group; low-dose HC + HO: high-cholesterol diet enriched with a low dose (5%) of HO; high-dose HC + HO: high-cholesterol diet enriched with a high dose (10%) of HO. GOT: aspartate aminotransferase; GPT: alanine aminotransferase; WK: week. Comparisons between groups were made using Duncan’s multiple range test. Superscript letters (^a–c^) indicate significant difference between groups (*p* < 0.05).

**Table 6 nutrients-11-02224-t006:** Liver pathological score of each group.

Histopathological Lesions	Group			
	Control	HC	Low-Dose HC + HO	High-Dose HC + HO
Fatty change, microvesicular, hepatocyte, diffuse, moderate/severe to severe/high	0 ± 0	4.9 ± 0.3 ^a^	4.9 ± 0.3 ^a^	4.4 ± 0.5 ^b^
Fatty change, macrovesicular, hepatocyte, multiple, minimal to moderate	0 ± 0	1.0 ± 0.0 ^b,c^	1.0 ± 0.0 ^c^	1.6 ± 0.5 ^a,b^
Infiltration, mononuclear cell and neutrophil, multiple, minimal to slight	0 ± 0	2.2 ± 0.4 ^a^	1.9 ± 0.3 ^a,b^	1.6 ± 0.5 ^b,c^
Hyperplasia, Kupffer cell, multiple, minimal to moderate	0 ± 0	2.2 ± 0.4 ^a^	2.0 ± 0.5 ^a,b^	2.0 ± 0.0 ^a,b^
Hyperplasia, oval cell, multiple, minimal to slight	0 ± 0	2.1 ± 0.3 ^a^	1.7 ± 0.5 ^b^	1.2 ± 0.4 ^c^
Fibrosis, multiple, minimal to slight	0 ± 0	2.1 ± 0.3 ^a^	1.7 ± 0.5 ^b^	1.2 ± 0.4 ^c^
Accumulation, glycogen, multiple, minimal to moderate	3.2 ± 1.55	2.1 ± 0.3 ^a^	1.1 ± 0.3 ^b^	1.9 ± 0.7 ^a^
Necrosis, hepatocyte, focal, minimal	0 ± 0	1.0 ± 0.0 ^a^	1.0 ± 0.0 ^a^	1.0 ± 0.0 ^a^

Control: basic diet control group; HC: high-cholesterol diet group; low-dose HC + HO: high-cholesterol diet enriched with a low dose (5%) of HO; high-dose HC + HO: high-cholesterol diet enriched with a high dose (10%) of HO. Superscript letters (^a–c^) indicate that a significant difference was detected by one-way ANOVA with Duncan’s multiple range test.

**Table 7 nutrients-11-02224-t007:** Hepatic and fecal lipid of each group.

Group	Hepatic Lipid (mg/g Liver)		Fecal Lipid (mg/g Feces)	
	TC	TG	TC	TG
**Control**	2.49 ± 1.19	1.69 ± 0.44	1.62 ± 0.04 ^a^	1.35 ± 0.31 ^a,b^
**HC**	13.59 ± 2.91 ^a^	3.26 ± 0.85 ^c^	1.57 ± 0.76 ^a^	1.08 ± 0.41 ^b^
**Low-dose HC + HO**	13.51 ± 3.59 ^a^	5.69 ± 1.53 ^b^	3.33 ± 2.41 ^a^	2.05 ± 0.18 ^a^
**High-dose HC + HO**	11.87 ± 4.10 ^a^	7.49 ± 2.70 ^a,b^	1.70 ± 0.67 ^a^	1.25 ± 0.70 ^a,b^

Control: basic diet control group; HC: high-cholesterol diet group; low-dose HC + HO: high-cholesterol diet enriched with a low dose (5%) of HO; high-dose HC + HO: high-cholesterol diet enriched with a high dose (10%) of HO; TC: total cholesterol; TG: total triglyceride. *n* = 10 in each group. Letters (^a–c^) indicate that a significant difference was detected by one-way ANOVA with Duncan’s multiple range test.

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
