# Peer review of "Dietary Supplementation with Hazelnut Oil Reduces Serum Hyperlipidemia and Ameliorates the Progression of Nonalcoholic Fatty Liver Disease in Hamsters Fed a High-Cholesterol Diet"

_nutrients, 2019, doi:10.3390/nu11092224_

Round 1
Reviewer 1 Report
The present study could be of interest to the readers of Nutrients.
I want to congratulation to the Authors a good manuscript.
Searching for new, natural substances helpful in treatment is very desirable, hence I am glad that the Authors have taken up this research topic. Especially, that PubMed presents only a few studies with supplementation on hazelnut oil.
However, there is still work to be done and some points need to be clarified. For this reason, my final decision is: minor revision.
The abbreviations GOT, GPT used in line 119 are not explained The abbreviation SGOT and SGPT used in line 198 are not explained In line 150 used μm, while in 112 verse was used mm All Tables are illegible, data difficult to read In chapter 2.5 (Determination of serum lipid and lipoprotein cholesterol), Authors do not write about VLDL-C measure, while in Results chapter this parameter is shown Table 4 - I can not understand why the hamster from Low-dose HO+HC groups had (in an initial moment) so the low concentration of TG, and so high concentration of LDL-C?! If the differences were statistically significant, that will mean hamster allocation was not correct. The results may be due to the initial state of the animals (it should be included in the discussion as a limitation of the study!) It seems to me that the description under Fig. 1 need not be as detailed (duplicating the same information like in main text) In line 284 should be ....TC and TG... (now is....TG and TG...) Literature numbering should be correctedAuthor Response
line 121:
GOT (Glutamic- Oxaloacetic Transaminase) and GPT (Glutamic Pyruvic Transaminase).
Line 115: should be 4 μm
We made improvement of our skill to paste table to the manuscript, hopefully it is now easier to read. We did not measure the VLDL and the VLDL-C value was calculated by formula (s. line 126): VLDL-C= TC- (LDL-C +HDL-C). line 382-386:Limitation and future works Compare to other groups, the animals from Low-dose HO+HC group had a significant low concentration of TG and high concentration of LDL-C at the initial stage, which is probably due to the wide normal range of serum lipid profile in hamster. Although the animals in Low-dose HO+HC group have successfully induced hyperlipidemia after feeding with an HC diet for 4 weeks at starting point (5th week) as other groups, this should be a limitation of this experiment.
The legend of Figure 1 is simplified as in line 222-226.
TC and TG in line 285 is corrected Literature numbering is now revised by endnote.

Reviewer 2 Report
This manuscript evaluated the effect of hazelnut oil supplementation on serum lipid profile and liver disease. In general, the results are nice. However, only hazelnut oil group was not included. Also, other oils, such as soybean oil and olive oil better be compared.
This study evaluated hazelnut oil supplementation on serum lipid profile and fatty liver disease in hamsters.
In general, the experimental results are nice.
However, control + HO(hazelnut oil) group better be included to see the effect of HO itself on the parameters studies in this study.
Also, some other oils such as soybean or olive oil better be compared because soybean oil is rich in polyunsaturated fatty acids and olive oil is rich in monounsaturated fatty acid.
Otherwise, positive control better be compared.
Author Response
“line 391-393”
Limitation and future worksThis experiment focused on the efficacy of dietary supplementation with different dose of HO with HC diet to reduce serum hyperlipidemia and fatty liver disease in hamsters. The effect of HO supplementation with normal diet on blood lipids and fatty liver disease remains unclear. Further study using different dose of HO with normal diet might reveal the effect of HO itself on the normalization of serum lipid profile and prevention NAFLD to NASH
Line 394-397
Limitation and future worksThe efficacy of HO can be further evaluated by comparison study with other dietary oils such as olive oil (rich in MUFA) and soybean oil (rich in PUFA).
